**Funding:** Grant awarded to SM Grant number-2020/1026382-0 Funder-World Health Organization URL of the funder website-https://

# Characteristics, experiences and actions taken by women to address delayed conception: A mixed-methods cross-sectional study protocol

Priyanka Adhikary[1], Nivedita Roy[1], Gitau Mburu[2], Rita Kabra[2], Ndema Abu Habib[2], James Kiarie[2], Neeta Dhabhai[1], Ranadip Chowdhury[1], Sarmila Mazumder[1]*

1 Centre for Health Research and Development, Society for Applied Studies, New Delhi, India, 2 UNDP-UNFPA-UNICEF-WHO-World Bank Special Programme of Research, Development and Research Training in Human Reproduction (HRP), Department of Sexual and Reproductive Health and Research, World Health Organization, Geneva, Switzerland

* sarmila.mazumder@sas.org.in

## Abstract

### Background

The impact of infertility on mental, emotional, physical and sexual health is grave, particularly in a pronatalist society. Literature is replete with evidence of wide ranging psychosocial consequences of infertility in women, indicating the need for identifying the gaps and designing appropriate context specific interventions to improve access and utilization of services. Data that are accessible, primarily from infertility clinics and women visiting hospitals for infertility treatment; information from community settings is rare. This is a protocol paper for a study to understand women's experiences and actions taken by them to cope with delayed conception.

### Methods

Mixed-methods cross-sectional design is used to obtain deep insights into the experiences of delayed conception, coping mechanisms, medical assistance and other help sought. Information is also being obtained on socio-demographic profile, fertility intentions, fertility quality of life, general medical history, obstetric, gynecological and sexual history, substance use and mental health status. A sample of 1530 women will be administered 4 modules of a quantitative survey. Focus group discussions, about four or till saturation point, will be conducted using purposive sampling. The study is recruiting from a population of women who previously participated in the 'Women and Infants Integrated Interventions for Growth Study (WINGS) and failed to conceive during 18 months follow up period. Data collected through questionnaire will be assembled, cleaned, analyzed and reported. The findings will be disseminated through reports with the ethics review committee, government entities, academic and research publications.

www.who.int/ The funders had and will not have a role in study design, data collection and analysis, decision to publish, or preparation of the manuscript.

**Competing interests:** The authors have declared that no competing interests exist.

**Abbreviations:** FGD, Focus-Group Discussion; WINGS, Women and Infant Integrated Interventions for Growth Study; FertiQo, Fertility Quality of Life; GHSQ, General Help Seeking Behavior; PHQ, Patient Health Questionnaire; QUANT-QUAL, Quantitative-Qualitative; STI, Sexually Transmitted Infections.

## Discussion

This study will provide insights on the experiences and coping strategies of women with delayed conception in the study community. Results will assist in designing appropriate interventions to meet the holistic health and psychosocial needs of women with delayed conception and promote sexual and reproductive health within the broader framework of Sustainable Development Goals and Universal health coverage.

## Trial registration

Trial registration number: CTRI/2020/03/023955.

## Background

The World Health Organization defines infertility as a disease of the male or female reproductive system defined by the failure to achieve a pregnancy after 12 months or more of regular unprotected sexual intercourse [1]. Globally, up to 16.7% of couples of reproductive age experience difficulties in conceiving [2, 3] and in India, it varies between 3.9 to 16.8% [4, 5]. Infertility or delayed conception has important impact on mental, emotional physical and sexual health, particularly in settings where bearing a child holds high value. In these settings, the ability to bear a biological child is considered an indication of a woman's health, and her position in society [6].

Literature is replete with evidence that infertility is associated with a wide-range of psychosocial consequences in women, including poor quality of life and marital relationships, depression, anxiety, decreased self-esteem, victims of violence, irrational thoughts, feelings of apprehensions and rejection, social stigma, somatization disorders, sexual dysfunction, and many more [6–12]. Significant association of psychosocial wellbeing has been found with age, occupation, education, and lack of social support [13, 14]. However, a majority of the existing studies are mostly from infertility clinics and hospital settings, where women seek care for infertility treatment. Studies from the community are rare.

An interesting paradox from global literature is that while childbearing is seen as important, 40–50% of infertile women in several studies did not seek help, often due to lack of cognitive awareness of the problem, or absence of symptom perception [15, 16]. Furthermore, in many low-and middle-income countries, infertility may not be recognized as a serious health problem that requires a combination of medical and psychological management [17]. Sub-optimal help seeking behavior is also frequently affected by prevailing attitudes towards informal and formal sources of fertility help [18–20] including costs associated with such help [16]. For instance, medical treatments such as assisted reproductive technologies are often unaffordable [21, 22]. Evidence from other settings suggest that medical treatment is the last resort, as women often first seek alternative treatment such as herbalists, spiritualists, and religious leaders as these are culturally more acceptable [21–23]. To date however, relatively limited research has examined how women cope with infertility in India [24, 25].

Understanding how many women experience infertility and how they cope with their circumstances is an essential first step towards providing them with necessary supportive interventions. Likewise, understanding household out-of-pocket expenditures that women and couples spend in their quest to conceive is important in formulating the kind of subsidies that

can be provided to such families in the context of fertility care services. In India, there is limited access to fertility and assisted reproduction services [26].

The present study, therefore, aims to describe the sociodemographic and other baseline characteristics of women who experience delayed conception, their perspectives, actions and experiences related to delay in conceiving, their quality of life and mental health and the costs of any interventions received.

## Methods

### Study design

This is a cross-sectional mixed methods study combining both quantitative and qualitative research methods. Close-ended objective questionnaire for the quantitative component and focus group discussions (FGDs) guide, for the qualitative component, are used for data collection.

### Study participants

Study participants are being recruited from a population of women who previously participated in the 'Women and Infants Integrated Interventions for Growth Study (WINGS) and did not become pregnant during the 18 months follow up period. WINGS is being conducted in the urban low-middle income neighborhood located in the South district of Delhi [27]. The purpose of WINGS is to enable women of reproductive age to enter pregnancy in good health, free of sexually transmitted infections (STIs), and well-nourished, through the implementation of an intervention package comprising nutrition, health, WASH and psychosocial care. The current study is an independent study where women are being recruited after they exit from WINGS. Based on current definitions, women of reproductive age (15–49 years) at risk of becoming pregnant (not pregnant, sexually active, not using contraception and not lactating) who report trying unsuccessfully for pregnancy for one year or more are classified as clinically infertile [28].

### Ethics approval and consent to participate

Approval for this study (Protocol version: 4.0 dated October 19, 2020) has been obtained from the Ethics Review Committee of the Centre for Health Research and Development, Society for Applied Studies (Approval number- SAS/ERC/RHR-Infertility/2020, approval date- January 9, 2020) and the World Health Organization, Geneva (Approval number- A-ID: A65998, approval date: February 13, 2020).

At the time of obtaining written informed consent, the research assistants, provide detail information about the new study. Women are asked questions to confirm that their participation in WINGS was completed at least 14 days earlier, their intention to become pregnant during their participation and after their exit in WINGS, and their interest to participate in the study on delay in conceiving. To ensure that women are not be under duress to consent to this study, clear statements are made to explain to the women that their participation is not obligatory, and enrolment is entirely voluntary. The study team comprises of well-trained personnel, with the required experience and expertise. In addition, the study team is trained in Good Clinical Practice and how to appropriately administer the consent form. The research assistant checks that all participants understand the above. At exit from WINGS, the WINGS study team seeks permission from the WINGS participants to share their contact details with the study team of the infertility study. If the participants give consent, only then the infertility study team visits the women at least 14 days after their exit from WINGS, to introduce the

study. Written individual informed consent is obtained during participants' enrolment. Consent is obtained for quantitative data collection and for FGDs. For those who are unable to read, the consent form is read out by the worker obtaining consent. In those who are unable to sign, a thumb imprint is taken which is witnessed (counter signed) by an impartial literate witness. The informed consent form is translated into simple Hindi language that can be easily read and understood.

The information collected from the study participants is kept confidential. The participants are not identified by her name but only by a study number. All information collected is stored in a locked area with access only to the study team. All study documents are be stored for a period of 5 years and will be destroyed at the end of the 5 years. The knowledge which we get from doing this research study will be shared through reports given to the ethics review committee, government agencies and publications but none will have the name of the study participants. No personally identifiable data is collected in this study. Only the unique study codes allocated to participants at enrolment are used in the questionnaires and focus group discussions. All data will have been aggregated before reporting. Furthermore, data is stored in-country by SAS and only reviewed by the immediate research team. All study team members are trained in Good Clinical Practice and are well versed with the principles of ethics. Women's autonomy is of prime importance, and we ensure that this is respected.

Women are provided with a small token of appreciation not exceeding 1.5 USD, alongside a free counseling on infertility. It is considered reasonable to compensate participants for their time and not an inducement.

This study enrolls women who had participated in WINGS, which has a large component of community participation. A significant part of contacts with participants is conducted at their homes, and local communities are continually reached and mobilized with the results to enhance uptake of interventions that promote better infant outcomes.

## Participant recruitment

At the time of exit from WINGS, consent for sharing the contact details of women with the new study team is sought, mentioning that this is a separate study and not a part of WINGS. If women agree with sharing of their contact details, the new study team, which is a separate team from WINGS, contacts these women at least 14 days after their exit from WINGS, to introduce the study. Women are contacted consecutively by field workers according to their exit dates, based on completion of 18 months' post-enrolment without getting pregnant. Women who agree to participate are invited to participate in the current study and at this point, written informed consent form is administered.

The study team screens women for eligibility. Women with delayed conception, living with husband, wishing to conceive and have not been using any form of contraceptives, are eligible to be enrolled (Fig 1).

## Sample size

For the purpose of the quantitative component of the study, the minimum sample size needed to estimate infertility or delayed conception, with 2% absolute precision and 95% CI from this sample would be 1223, assuming a prevalence of 17% as reported in the literature [4, 5], and a finite population of 12,500. An allowance of 25% for non-responses has been made. Final sample size is therefore 1530. The study enrolment will continue on a consecutive basis until the required 1530 women who did not become pregnant at 18 months are interviewed.

A purposive sampling is drawn to conduct FGD from women who participate in the quantitative survey. Recruitment will be informed by saturation of data, or to a maximum of 50

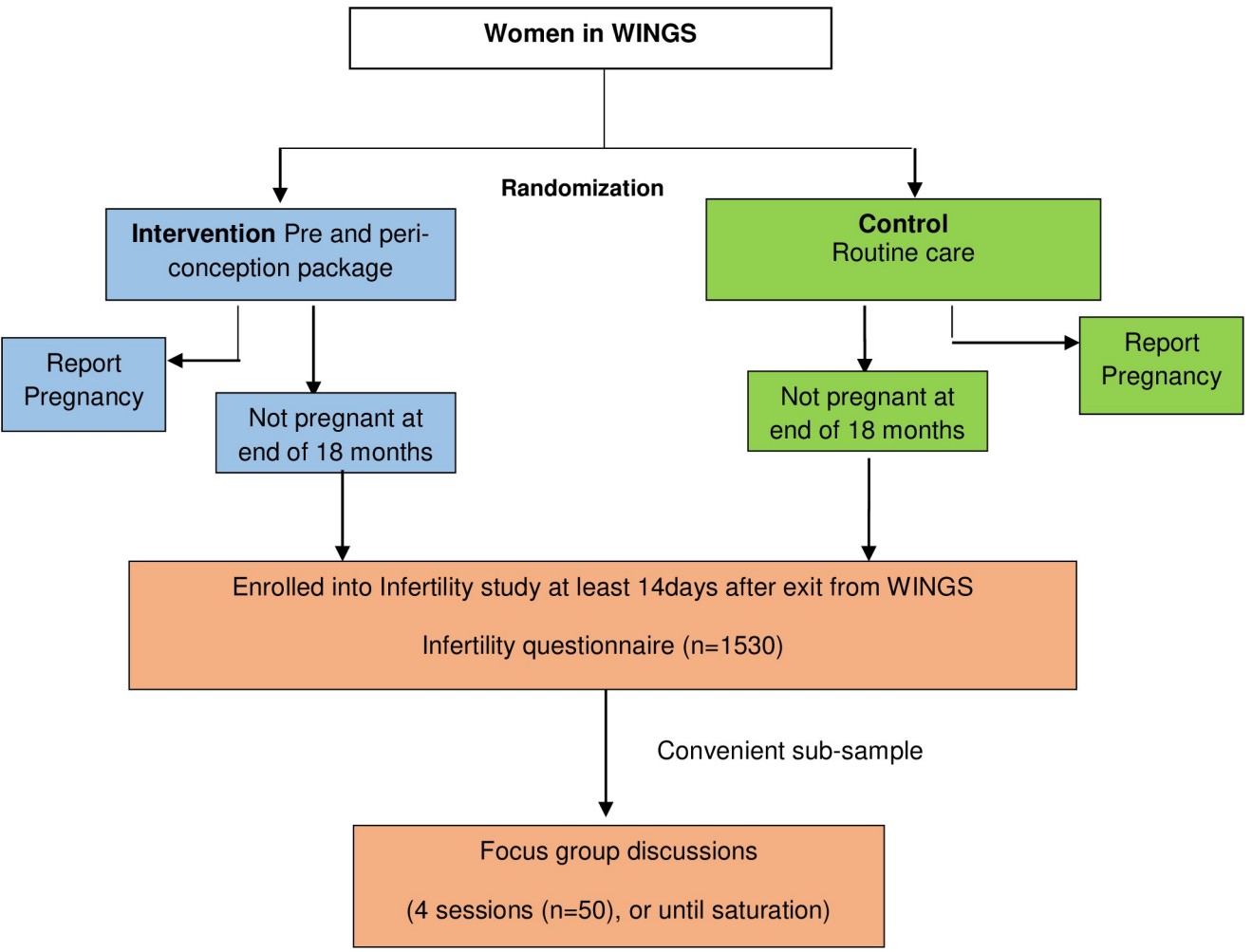

**Fig 1. Overview of the study design.**

participants, approximately 4 sessions of 10–15 participants each. Data will reach a saturation point when no new information will be generated during the FGDs and further data collection will not yield additional findings. FGD audio recordings will be used to assess whether the FGDs no longer bring out any new information or knowledge to confirm data saturation. The intent in purposive sampling is to obtain a representative of the sample of a more homogenous group (experienced primary or secondary infertility or delayed conception) than the total population.

## Survey procedure

Once informed written consent is obtained, face-to-face interviews are conducted with the women, using a standardized survey questionnaire. The questionnaire comprises the following four modules/sections:

Module 1 comprises 20 background questions on social demographic variables and fertility intentions. These questions have been adopted from previous surveys and studies related to infertility. They capture information related to age of partners, duration of infertility and employment among others.

Module 2 comprises the Fertility Quality of Life (FertiQoL) questionnaire (English and a translated Hindi version). The FertiQoL questionnaire is a validated instrument developed by

an international collaboration of experts, for specifically assessing quality of life in all people experiencing fertility problems [29]. As opposed to generic measure of quality of life, it is validated and specific to infertility (www.fertiqol.org). FertiQoL assesses the impact of infertility on four domains that is Mind-Body (six items), Relational (six items), Social (six items) and Emotional (six items). It assesses these impacts on different formats of a five-point Likert scale: (i) from very poor to very good (one item); (ii) from very dissatisfied to very satisfied (seven items); (iii) from completely to not at all (four items); (iv) from always to never (eight items) and (v) from an extreme amount, to not at all (six items). FertiQoL comprises two modules: a core-FertiQoL module and an (optional) treatment-module. For this study the treatment module was excluded as the study was not formally providing treatment for infertility or delayed conception. This is particularly salient given the emphasis by researchers regarding the importance of making use of non-clinic-based samples to better understand the experience of infertility [30]. The final FertiQoL versions, including the Hindi version used in this study have all been professionally translated and checked by fertility experts to ensure appropriateness of wording to local customs and usage, including in India.

Module 3 comprises 13 questions on actions taken including help seeking. Actions taken by women are assessed in three key domains: the first is help seeking, using format adapted from the General Help Seeking Questionnaire (GHSQ) [31]. The GHSQ was originally developed to assess future help seeking from a list of culturally-relevant sources [31], but may be used to identify actual help seeking. The sources of help included in the GHSQ are classified into two major classes: formal (such as doctors, traditional and faith healers), and informal (such as parents, friends, a partner or other relatives). We adapted the GHSQ to ask about actual (rather than intended) behaviors, and adapted the sources as locally relevant, and collapsed the response categories to YES/NO as have been done in other studies [32]. We also include a category "did not seek help" so that we can identify participants that did not seek help. The second domain to be assessed regarding women's action is their out of pocket expenditure on the above sources of help by asking women to estimate the amounts of money they spent on each of the above sources of help (which may or may not be catastrophic). To assess barrier to access to care, the questionnaire assess the importance of 24 barriers to access to care items validated from other studies [33], which were piloted and modified based locally relevant barriers. Finally, the survey collects data on adoption of practices such as vaginal douching, use of traditional remedies, seeking, medical assistance, or other changes in sexual behavior and limited information on action taken by husbands, care seeking by husbands, or any sexual problem faced by husbands.

Module 4 comprises questions exploring general medical history including obstetric, gynecological, sexual, substance use and mental health. The module collects data on social demographic variables, history of hormonal or other co-morbidities, detailed menstrual cycle history, abortions, whether spontaneous or medically terminated, STI and sexual behavior, alcoholism, and smoking. A validated 4-item Patient Health Questionnaire-4 (PHQ-4) is used to assess mental health (depression and anxiety). PHQ-4 is an ultra-brief self-report questionnaire that consists of a 2-item depression scale (PHQ-2) and a 2-item anxiety scale (GAD-2) [34].

It takes 1.5 to 2 hours to respond to all the four modules, constituting an estimated 100 questions. During piloting the study team had assessed the necessity to conduct the interviews in more than 2 rounds/sessions, for example if these questions are overwhelming, or if the time required responding is too long. Piloting of the questionnaire also assessed the interpretation and use of alternative terms to infertility, such as "difficulties in conceiving", "delay in conceiving", or "failure to conceive" to ensure relevance and cultural sensitivity.

Risk of infertility is increased due to STI. STI symptoms are identified as part of the questionnaire under the sexual health section. Study participants identified with symptoms of

vaginal discharge are examined at the study clinic by study physician (MD Obstetrics and Gynaecology), and provided treatment free of cost, based on the syndromic approach, as recommended in the Government of India guidelines. If required, referral of the participant and/ or her husband, to the collaborating tertiary care hospital, Safdarjung hospital, is facilitated by the study team, for further evaluation and management. Regular infertility clinics are conducted in the hospital. Investigations and treatment are free of cost in the collaborating hospital. Transportation arrangements are made by the study team.

At the end of questionnaire administration, women are offered free counselling regarding their delay in conceiving by the gynaecologist and obstetrician. If the women want, they are also referred to the infertility clinic at Safdarjung hospital. For any emotional distress pertaining to infertility, the study psychologist conduct free counselling sessions with the woman, her husband and other family members. The study team may become aware of interpersonal violence from their partners/husbands. The study does not provide direct intervention in matters of domestic violence or abuse, which may worsen the situation further. We have identified organizations locally that offer help and support to women who are victims of domestic and intimate partner violence/abuse. If women are comfortable and willing to seek such help, they are provided with the linkages and offered assistance, in terms of transportation to access these services.

Women responding to the quantitative questionnaire are informed and invited to participate in FGDs and based on their interest and availability, included in the FGDs. This is a purposive sample drawn from women who participate in the survey. These FGDs provide in-depth information regarding women's and to some limited extent, their husbands' perceptions, experiences, actions and motivations for such actions, while contextualizing the quantitative data. The use of qualitative methods in exploring experiences of infertility is critical given that infertility is a socially constructed phenomenon [35].

FGDs are guided by topic guide covering a range of topics such as experiences of delayed conception, coping with delayed conception, vaginal douching, traditional remedies, help seeking, including specialized medical assistance, and other actions. A field trial of the FGD topic guide was conducted on a small sample of women (n = 6) and revisions were made based on feedback to ensure relevance and cultural sensitivity. FGDs are conducted following the questionnaire, in a QUANT-QUAL sequence [36]. FGDs are audio recorded and held at a WINGS site in local language. These last for 45–60 minutes.

## Data analysis

The quantitative data will be assessed for completeness and cleaned. To meet objective 1, characteristics of women participate in this study and who exit the WINGS at 18 months without becoming pregnant will be described using descriptive statistics. These characteristics include social demographic, gynecological, obstetric, family, mental health and substance use as described in the questionnaire. To meet objective 2, on quality of life of women participating in this sub-study, scores of FertQual questionnaire will be derived.

Overall scores and sub-scores will be calculated for both the FertiQoL following the established scoring criteria for each questionnaire. FertiQoL will be scored to derive mean values (a value of 0 indicates the most negative answer and a value of 4 indicates most positive). The scores for each question will be from very poor (= 0) to very good (= 4), from very dissatisfied (= 0) to very satisfied (= 4), from completely (= 0) to not at all (= 4), from always (= 0) to never (= 4) and from an extreme amount (= 0) to not at all (= 4). After that, the interpretation of FertiQoL total and subscale scores will range from 0to 100. The consistency of FertiQoL questionnaire will be assessed by computing a Cronbach's α coefficient of these scales. Means, standard

deviations (SDs) and proportions will be calculated, and parametric tests were used on variables for medical history in the study.

Actions and barriers to seeking help will be analyzed by deriving proportions of women who report specific barriers. Exploratory analysis will be conducted to analyze predictors of help seeking. These will be complemented with mental health scores from PHQ4. Assessments of depression and anxiety will be conducted by assessing participants' responses to the questions, "over the last 2 weeks, how often have you had "little interest or pleasure in doing things" and "over the last 2 weeks, how often have you been "feeling down, depressed, or hopeless" and 'over the feeling nervous, anxious or on edge", "over the last 2 weeks, how often have you "not being able to stop or control worrying?" Participants will be required to select/check only one answer from four options i.e., "not at all", "several days", "more than half the days or nearly every day."

Experiences and actions taken by women will be described using data from both the quantitative survey and the qualitative FGDs. These will include experiences of violence, stigma, divorce, separation as well as help-seeking and the cost of assistance sought in relation to infertility. Qualitative analysis will document information related to experiences and subsequent actions of delayed conception, coping with delayed conception, vaginal douching, traditional remedies, help seeking (from traditional, family, religious, or medical sources) as well as and socio-economic impacts of delayed conception.

Qualitative analysis will be conducted via an inductive process [37] in which nodes and codes will be developed in Nvivo [38] and semantic themes derived based on most commonly occurring phenomena, related to women's experiences and actions and motivations for those actions.

Interpretative stance will be assumed, and an intersectionality approach will be used to elaborate ways in which both gender and other axes of social locations intersect to determine experiences of women, and how differential and dynamic experiences may occur based on the varying nature of participants social, economic or positions in society. Given that gender roles has an impact on women's experiences of infertility in Asia [39], it is expected that intersectionality would be useful in explaining varying ways in which gender intersects with other axes, such as economic deprivation- to produce different experiences among women [39]. To elaborate specifically on the actions that women take, the theory of planned behavior, which is an extension of the Theory of Reasoned Action [40] will be drawn upon. The use of these two theoretical frameworks will add credibility and rigor to the analysis. Following the development of descriptive themes, analytical themes will then be derived with reference to the literature and the study context. Further hypothesis may be developed following the above exploratory analysis.

## Discussion

This study aims to better understand the sexual and reproductive contexts of women through exploring the characteristics, psychosocial impact and actions taken by women with delay in conceiving. Infertility has been widely studied from clinical perspective but due to the sensitive nature of the topic, previous studies failed to capture accurate and relevant information about the effect of infertility on marital and sexual life in India [41]. There is also a dearth of community based studies, most of the literature reporting the negative impact of infertility on mental health and quality of life, are from hospital settings and include women who actively seek care. The present study enrolls women from the community, including those women who may not have sought help. Additionally, the familiar home environment allows women to be comfortable and feel free to share their perspectives and experiences.

Apart from assessing the quality of life, psycho-social impact of infertility, and actions taken by women if any, when conception is delayed, this study will provide useful information on potential barriers in accessing appropriate care, assess the need for intervention, and identify potential interventions. The FGDs will play a pivotal role in encouraging women to express their feelings, experiences and distress and provide a platform for sharing and interacting with others, thus providing deep insights and identify the need for urgent action, with suitable interventions, including social support.

This study does not involve the husbands and other family members of the participants or the health care providers in the area catering to problems related to delayed conception, hence may miss relevant additional perspectives and information which may be useful in further contextualizing the findings. This study samples participants from a population of women who previously participated in a study in Northern part of India, which may limit generalizability.

## Conclusion

This study will contribute to better understanding of the characteristics, experiences, and coping strategies of women with delayed conception in the study community. Results will assist in designing appropriate interventions to meet the holistic health and psychosocial needs of women with delayed conception and promote sexual and reproductive health within the broader framework of Sustainable Development Goals, and universal health coverage. The dissemination of study findings will be done through presentations and publications at scientific meetings and conferences, and peer reviewed journals respectively. The results will also be shared with relevant professional bodies and non-governmental organizations at national and sub-national levels.

## Supporting information

**S1 File. CHRD SAS ERC infertility approval letter.**
(PDF)

**S2 File. WHO ERC infertility approval letter.**
(PDF)

**S3 File. Consent form.**
(PDF)

**S1 Questionnaire. Questionnaire modules.**
(PDF)

**S1 Checklist. SPIRIT 2013 checklist: Recommended items to address in a clinical trial protocol and related documents\*.**
(DOC)

## Acknowledgments

The authors are thankful to the implementation partners of WINGS including CHRD-SAS, Safdarjung Hospital, and the Department of Maternal, Newborn, Child and Adolescent Health, WHO. The authors also extend their gratitude to the participants of the study. The views expressed are those of the authors and not necessarily those of their respective institutions.

## Author Contributions

**Conceptualization:** Gitau Mburu, Rita Kabra, Ndema Abu Habib, James Kiarie, Neeta Dhabhai, Ranadip Chowdhury, Sarmila Mazumder.

**Funding acquisition:** Sarmila Mazumder.

**Supervision:** Priyanka Adhikary, Nivedita Roy.

**Writing – original draft:** Priyanka Adhikary, Nivedita Roy, Sarmila Mazumder.

**Writing – review & editing:** Priyanka Adhikary, Nivedita Roy, Gitau Mburu, Rita Kabra, Ndema Abu Habib, James Kiarie, Neeta Dhabhai, Ranadip Chowdhury, Sarmila Mazumder.

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
