## [Decision Letter · Decision Letter 0]

27 Jan 2022

PONE-D-21-37261Characteristics, experiences and actions taken by women to address delayed conception: A mixed-methods cross-sectional study protocolPLOS ONE

Dear Dr. Mazumder,

Thank you for submitting your manuscript to PLOS ONE. After careful consideration, we feel that it has merit but does not fully meet PLOS ONE’s publication criteria as it currently stands. Therefore, we invite you to submit a revised version of the manuscript that addresses the points raised during the review process.

Two reviewers have gone through the manuscript finding that it is amenable for publication with minor revisions and including more information, in particular the questionnaire.

We look forward to receiving your revised manuscript.

Kind regards,

José Antonio Ortega, Ph.D.

Academic Editor

PLOS ONE

Journal Requirements:

Reviewers' comments:

Reviewer's Responses to Questions

**Comments to the Author**

1. Does the manuscript provide a valid rationale for the proposed study, with clearly identified and justified research questions?

Reviewer #1: Yes

Reviewer #2: Yes

2. Is the protocol technically sound and planned in a manner that will lead to a meaningful outcome and allow testing the stated hypotheses?

Reviewer #1: Yes

Reviewer #2: Yes

3. Is the methodology feasible and described in sufficient detail to allow the work to be replicable?

Reviewer #1: Yes

Reviewer #2: Yes

4. Have the authors described where all data underlying the findings will be made available when the study is complete?

Reviewer #1: Yes

Reviewer #2: Yes

5. Is the manuscript presented in an intelligible fashion and written in standard English?

Reviewer #1: Yes

Reviewer #2: Yes

6. Review Comments to the Author

You may also provide optional suggestions and comments to authors that they might find helpful in planning their study.

Reviewer #1: Well planned study design. The questionnaire is too long and may not be accepted by patients. Minor grammatical errors need correction.

Reviewer #2: This is a very that well-written and compelling study protocol.

I have only a few recommendations as described below:

Abstract

Line 32: It is not clear what “four” is referring to. Did you mean to state “four sessions”?

Background

Line 69: It seems more appropriate not label women with the term “infertile women” but rather describe them as “women with infertility.

Line 74: I would recommend changing “fertility help” to fertility assistance.

Line 88: Please enter a comma after “mental health”

7. PLOS authors have the option to publish the peer review history of their article (what does this mean?). If published, this will include your full peer review and any attached files.

Reviewer #1: No

Reviewer #2: No

---

## [Author Response · Author response to Decision Letter 0]

16 Feb 2022

We have gone through the templates and made required changes to meet the journal requirements.

2a. The data in the study are sensitive in nature. The contact details of the ethics review committee is given below:

Ethics Review Committee, Centre for Health Research and Development, Society for Applied Studies, 45, Kalu Sarai, New Delhi- 110016, India; Contact numbers: +919582595320; +917838350052; email ID: chrderc@sas.org.in

2b. This is not applicable as this is a protocol paper. 

Supporting information files have been mentioned at the end of the manuscript including captions as desired. 

4. Please review your reference list to ensure that it is complete and correct. If you have cited papers that have been retracted, please include the rationale for doing so in the manuscript text or remove these references and replace them with relevant current references. Any changes to the reference list should be mentioned in the rebuttal letter that accompanies your revised manuscript. If you need to cite a retracted article, indicate the article’s retracted status in the References list and also include a citation and full reference for the retraction notice.

We have reviewed the references, these are complete and correct. To the best of our knowledge, none of the cited papers has been retracted. 

Comments to the author 

Reviewer #1: Well planned study design. The questionnaire is too long and may not be accepted by patients. Minor grammatical errors need correction.

Thank you very much for you comment, we agree with you. We have pretested the questionnaire for feasibility. Since we have a good rapport with the community, women in the community like to interact and share their experiences with us. If we feel at any point that the participant is reluctant to continue with data collection or is tired or has other commitments to fulfil, we will stop immediately and request for another day and time to continue with the data collection process. This may require multiple visits. We will ensure that administration of the questionnaire is according to the participants’ convenience. 

Thank you for pointing out grammatical errors, we have corrected them. 

Minor grammatical errors need correction 

1. Abstract background line number 22 and 23- The sentence is not imparting a clear meaning as to what the author wants to convey. 

Thank you for your comment, we have modified the sentence for greater clarity from line number 25 to27 and page number 2

2. Line number 118 - Modify sentence “women are not be under duress”

Thank you for pointing this out, we have now corrected it. Line number 121and page number 6

3. Line number 134 - All study documents are to be stored (to added)

Thank you for pointing this out, we have now corrected it. Line number 137and page number 7

4. Detailed questionnaire is not attached

We have now attached the detailed questionnaire as a supplement. 

Reviewer #2: This is a very that well-written and compelling study protocol.

I have only a few recommendations as described below:

Abstract

Line 32: It is not clear what “four” is referring to. Did you mean to state “four sessions”?

Thank you for your comment. Four (4) refers to number of modules, each module having a set of questions that will be used in the study. The sentence has been modified for a clear understanding. Line number 34 and page number 2.

Background 

Line 69: It seems more appropriate not label women with the term “infertile women” but rather describe them as “women with infertility.

We have changed from infertile women to women with infertility. Thank you again for such valuable feedback. Line number 72 and page number 4.

Line 74: I would recommend changing “fertility help” to fertility assistance.

We really appreciate for such insight. We have modified to fertility assistance. Line number 77 and page number 4

Line 88: Please enter a comma after “mental health”

Done, thank you. Line number 91 and page number 5

We hope we have addressed your comments satisfactorily and the manuscript will be considered for publication in the journal.

---

## [Editor Report · Decision Letter 1]

17 Feb 2022

Characteristics, experiences and actions taken by women to address delayed conception: A mixed-methods cross-sectional study protocol

PONE-D-21-37261R1

Dear Dr. Mazumder,

We’re pleased to inform you that your manuscript has been judged scientifically suitable for publication and will be formally accepted for publication once it meets all outstanding technical requirements.

Kind regards,

José Antonio Ortega, Ph.D.

Academic Editor

PLOS ONE

Additional Editor Comments (optional):

The suggested changes have been addressed. Congratulations. Looking forward to the results.
---

## [Editor Report · Acceptance letter]

28 Feb 2022

PONE-D-21-37261R1 

Characteristics, experiences and actions taken by women to address delayed conception: A mixed-methods cross-sectional study protocol 

Dear Dr. Mazumder:

I'm pleased to inform you that your manuscript has been deemed suitable for publication in PLOS ONE. Congratulations! Your manuscript is now with our production department. 

Kind regards, 

on behalf of

Dr. José Antonio Ortega 

Academic Editor

PLOS ONE